# Unfolding States of Mind: A Dissociative-Psychedelic Model of Ketamine-Assisted Psychotherapy in Palliative Care

**DOI:** 10.3390/healthcare13212714

**Published:** 2025-10-27

**Authors:** Alessandro Gonçalves Campolina, Marco Aurélio Tuena de Oliveira

**Affiliations:** 1Centro de Investigação Translacional em Oncologia, Instituto do Câncer do Estado de São Paulo, Faculdade de Medicina FMUSP, Universidade de São Paulo, Sao Paulo 01246-903, SP, Brazil; 2Hospital Samaritano, Sao Paulo 01232-010, SP, Brazil; drmarcotuena@gmail.com

**Keywords:** ketamine, psychedelics, palliative care, ketamine-assisted psychotherapy cancer, chronic illness

## Abstract

**Background/Objectives**: Patients in palliative care often experience multifaceted forms of suffering that extend beyond physical symptoms, including existential distress, loss of meaning, and emotional pain. Ketamine-assisted psychotherapy (KAP) has emerged as a promising intervention for alleviating such complex forms of suffering, yet models specifically tailored to palliative populations remain scarce. This narrative review synthesizes current evidence on ketamine’s neurobiological, psychological, and experiential effects relevant to end-of-life care, and presents a novel, time-limited KAP model designed for use in palliative settings. **Methods**: Drawing from both biochemical and psychedelic paradigms, the review integrates findings from neuroscience, phenomenology, and clinical practice. In particular, it incorporates a dual-level experiential framework informed by recent models distinguishing ketamine’s differential effects on self-processing networks: the Salience Network (SN), related to embodied self-awareness, and the Default Mode Network (DMN), associated with narrative self-construction. This neurophenomenological perspective underpins the rationale for using two distinct dosing sessions. **Results**: The article proposes a short-course, time-limited KAP model that integrates preparatory and integrative psychotherapy, two ketamine dosing sessions (one low-dose and one moderate-dose), concurrent psychotherapy, goals of care discussion (GOCD), and optional pharmacological optimization. The model emphasizes psychological safety, meaning-making, and patient-centered care. The sequential dosing strategy leverages ketamine’s unique pharmacology and experiential profile to address both bodily and narrative dimensions of end-of-life distress. **Conclusions**: This dissociative-psychedelic model offers a compassionate, pragmatic, and theoretically grounded approach to relieving psychological and existential suffering in palliative care. By integrating neurobiological insights with psychotherapeutic processes, it provides a flexible and patient-centered framework for enhancing meaning, emotional resolution, and quality of life at the end of life. Further research is needed to evaluate its clinical feasibility, safety, and therapeutic efficacy.

## 1. Introduction

Over the past two decades, researchers, clinicians, and cultural commentators have increasingly revived interest in classical psychedelics and psychotomimetic substances—a movement commonly known as the Psychedelic Renaissance [1,2]. After decades of stigmatization and prohibition that began in the 1970s, substances such as psilocybin, LSD, ayahuasca, and mescaline—as well as dissociative compounds like ketamine—have been reintroduced into clinical and research settings under rigorous ethical and methodological protocols [2].

The first wave of clinical research with classical or serotonergic psychedelics took place between the late 1950s and the mid-1970s in Europe and the United States. During this period, more than 40,000 participants were involved—primarily in studies using LSD and, to a lesser extent, psilocybin—culminating in the publication of over a thousand scientific articles [3].

Recent studies have shown the therapeutic potential of these substances in the treatment of resistant mental health conditions such as major depressive disorder [4], post-traumatic stress disorder, anxiety related to terminal illness [5], and substance use disorders [6]. The subjective effects induced by psychedelics—often characterized by ego dissolution, emotional intensification, and cognitive reorganization—appear to be closely linked to their positive clinical outcomes [7].

Unlike conventional pharmacological approaches, psychedelic-assisted therapy emphasizes the role of “set and setting” (i.e., mental preparation and the physical and relational context), as well as the importance of integrating the insights and experiences gained during altered states of consciousness [8]. This signals a paradigmatic shift: psychedelics are not merely pharmacodynamic agents, but also catalysts for profound psychotherapeutic processes, whose efficacy depends on the interplay between biological, psychological, and existential dimensions.

In the current, modern era of psychedelic research, studies involving individuals affected by cancer have not reported any serious medical or psychiatric adverse events linked to the psychedelic compounds themselves. This suggests that, when administered with appropriate risk mitigation strategies—including the exclusion of individuals with psychotic disorders, thorough psychotherapeutic preparation, and continuous support during and after sessions—the therapeutic use of psychedelics in patients with serious illnesses can be considered safe [9].

Within oncology care, particularly at the end of life, there remains a persistent clinical gap in effectively addressing depression, anxiety, and existential distress. Pharmacological and psychological treatments for these symptoms in terminally ill patients have shown limited efficacy. While some antidepressants—such as mianserin, fluoxetine, and paroxetine—have demonstrated mild effects, response rates are typically low and often not significantly better than placebo. Data supporting the use of anxiolytics in patients with advanced cancer are even more limited [10].

On the non-pharmacological front, recent psychotherapeutic approaches developed to address existential suffering in terminal cancer patients have emphasized the importance of meaning, dignity, and psychological well-being. These interventions often yield significant but short-lived improvements in existential well-being, hope, and self-care, yet show limited impact on anxiety and depression. As such, existential distress, depression, and anxiety in patients with incurable illnesses continue to represent unmet clinical needs, pointing to the urgent demand for novel therapeutic alternatives [11].

In this context, the use of psychedelic compounds to ease existential distress at the end of life has regained interest, echoing a historical and philosophical appeal. This vision was perhaps most famously embodied by British author Aldous Huxley, who, on his deathbed in 1963, requested that his wife administer two doses of LSD. His passing was described as peaceful, guided by his wife’s presence and a metaphorical journey “toward the light” [12].

Among the substances gaining clinical prominence, ketamine occupies a unique position due to its legal status, rapid onset of action, and established safety profile. Ketamine is a dissociative anesthetic that was first synthesized in 1962 as a shorter-acting and more cardiorespiratorily stable analog of phencyclidine [13]. It is a racemic mixture composed of two enantiomers—molecules with identical compositions but differing spatial arrangements as mirror images of one another. These enantiomers exhibit distinct pharmacological properties, with (S)-ketamine, or esketamine, demonstrating a higher affinity for N-methyl-D-aspartate (NMDA) receptors, greater potency as an anesthetic and analgesic, and more pronounced psychotomimetic effects compared to (R)-ketamine [13].

Over the past two decades, ketamine—and more recently, its enantiomer esketamine—has gained recognition as an innovative pharmacological treatment, offering new hope for individuals with treatment-resistant depression and other mental health disorders [14]. Today, it is acknowledged as a therapeutic option in psychopharmacology literature and is considered on par with established treatments such as electroconvulsive therapy (ECT) and transcranial magnetic stimulation (TMS) [14,15].

In palliative care settings, where patients often face complex emotional, existential, and spiritual distress, ketamine-assisted psychotherapy (KAP) emerges as a promising intervention. Preliminary studies suggest that ketamine may alleviate suffering not only through neurochemical mechanisms, but also by facilitating non-ordinary states of consciousness that promote meaning-making, emotional processing, and psychosocial integration [16]. This article introduces a clinical model of KAP tailored for palliative care populations, emphasizing experiential depth, patient-centered preparation and integration, and the therapeutic use of music, setting, and expanded states of awareness.

## 2. Materials and Methods

This study followed a narrative review design aimed at providing an interpretative and integrative synthesis of the literature on KAP in palliative care. Guided by an interpretivist perspective, the review sought to describe current knowledge, theoretical frameworks, and clinical approaches related to KAP in end-of-life contexts.

An iterative and flexible search strategy was conducted across major scientific databases and relevant grey literature, focusing on studies, conceptual papers, and clinical reports addressing the therapeutic use of ketamine for psychological, existential, or spiritual distress in palliative care. Sources were selected based on conceptual and thematic relevance rather than strict inclusion criteria.

The analysis involved identifying recurrent themes, models, and gaps within the literature, integrating empirical findings with theoretical perspectives from psychotherapy, psychedelic science, and palliative medicine. Reflexivity was maintained throughout the process, recognizing the influence of the authors’ clinical and theoretical background on interpretation.

To lay the groundwork for our proposed KAP model for patients in palliative care, we begin by examining current understandings of ketamine’s mechanisms of action, both at the molecular level and in terms of its effects on large-scale neural networks. This neurobiological foundation informs a growing body of clinical evidence supporting ketamine’s use across a range of psychiatric conditions. We then contrast three prevailing models of clinical application—the biochemical, psychotherapeutic, and psychedelic—each of which reflects distinct assumptions about how ketamine works, how it should be delivered, and what therapeutic outcomes are prioritized. Thus, this analysis is based on a narrative/conceptual review of the scientific literature and provides essential context for the development of an integrative model that draws selectively from each tradition to address the complex psychological and existential needs of individuals in palliative care.

### 2.1. Mechanisms of Action

#### 2.1.1. NMDA Antagonism and Neurotransmitter Influence

There are two prominent, non-mutually exclusive hypotheses explaining ketamine’s antidepressant effects via NMDA antagonism: the Disinhibition Hypothesis and the Direct Inhibition Hypothesis [17,18,19,20,21].

Disinhibition Hypothesis: This hypothesis proposes that ketamine blocks NMDA receptors on GABAergic interneurons, reducing inhibitory control over pyramidal neurons. This disinhibition increases glutamate release in the medial prefrontal cortex (mPFC) and elevates the firing rate of pyramidal neurons. Such a “glutamate surge” promotes AMPA receptor activation relative to NMDA activation, enhancing synaptic activity and potentially contributing to antidepressant effects. Supporting this view, AMPA receptor antagonists or mGluR2/mGluR3 agonists can block ketamine’s behavioral effects, indicating the importance of elevated extracellular glutamate levels [17,18,19].Direct Inhibition Hypothesis: This hypothesis suggests that ketamine directly antagonizes NMDA receptors on resting pyramidal neurons, blocking tonic NMDA activation by ambient or spontaneously released glutamate. This blockade reduces suppression of protein synthesis and activates downstream synaptogenic cascades. Increased AMPA receptor activation through this mechanism is also thought to contribute to ketamine’s antidepressant effects [20,21].

Research findings are not entirely consistent. While some studies indicate that ketamine increases glutamate release in the PFC, others suggest it decreases glutamate levels in the cortex. However, ketamine consistently increases evoked excitatory postsynaptic potentials/currents in pyramidal neurons, similar to the effects observed with psilocybin [21,22].

In addition to its glutamatergic actions, ketamine modulates other neurotransmitter systems. It transiently increases extracellular serotonin, dopamine, and norepinephrine levels in the prefrontal cortex via glutamatergic projections to monoaminergic nuclei, which may contribute to improvements in mood and, notably, anhedonia [23]. Evidence also indicates that ketamine upregulates GABAA receptor activity and enhances GABA turnover, although excessive GABAergic potentiation (e.g., from benzodiazepines) appears to blunt its antidepressant efficacy.

Finally, ketamine interacts with the opioid system, showing moderate affinity for mu- and kappa-opioid receptors. Clinical studies have reported that opioid receptor antagonism can abolish ketamine’s antidepressant response, suggesting a modulatory role of opioid signaling [24]. Taken together, ketamine’s antidepressant profile emerges from the convergence of glutamatergic disinhibition, transient monoaminergic enhancement, GABAergic modulation, and opioid receptor interactions, distinguishing it mechanistically from conventional monoaminergic antidepressants.

#### 2.1.2. BDNF and mTOR Signaling

Ketamine induces structural and functional changes in pyramidal neurons, largely mediated through the enhancement of neuroplasticity-related pathways. Animal studies have shown that ketamine increases brain-derived neurotrophic factor (BDNF) levels in the cortex in a dose-dependent manner. Notably, ketamine’s antidepressant effects are absent in BDNF knock-out mice, underscoring BDNF’s critical role in its mechanism of action [25,26].

Upon release, BDNF binds to TrkB receptors and activates mTOR signaling through MEK–ERK and Akt pathways. Increased mTOR activity promotes synaptogenesis and the expression of neuroplasticity-related genes. Inhibition of mTOR abolishes ketamine’s synaptogenic and antidepressant effects in animal models [23,27,28].

#### 2.1.3. Structural and Functional Changes

Ketamine reduces suppression of protein synthesis mediated by eukaryotic elongation factor 2 (eEF2) and engages downstream synaptogenic cascades [29,30]. This supports dendritic complexity, spine density, and synaptic strength over extended periods, contributing to ketamine’s rapid and sustained antidepressant effects [19,20,21,22,23,24,25,26,27,28,29,30,31]. In rodents, ketamine reverses behavioral correlates of depression within 30 min, with effects lasting approximately one week [18,19,20,21,22,23,24,25,26,27,28,29,30,31,32].

#### 2.1.4. Functional Connectivity Changes

Ketamine administration has been shown to alter functional connectivity within and between various brain networks. Studies have found decreased connectivity within the Default Mode Network (DMN) and Salience Network (SN), and reduced anti-correlation between the DMN and other networks post-administration [33]. Increased connectivity between the Central Executive Network (CEN) and resting-state networks (DMN and SN) has also been reported [33].

Ketamine appears to decrease within-network functional connectivity and enhance between-network connectivity, resembling the effects of psilocybin. Studies using fMRI and EEG have reported decreased functional connectivity in the medial PFC, increased connectivity in the intraparietal cortex, and shifts to delta, gamma, and theta waves, with reduced alpha power [34].

In depressed patients, ketamine has been found to increase global signal regression (GBCr) in the lateral PFC, caudate, and insula 24 h post-infusion [31,35]. Increased GBCr in the PFC is positively correlated with improvements in depressive symptoms [35]. However, some studies have failed to replicate these findings, suggesting the effects of ketamine on functional connectivity may vary between individuals [36].

Additionally, ketamine has been shown to increase frontostriatal connectivity in MDD patients to levels observed in healthy controls, while altering connectivity profiles of healthy individuals to resemble those of treatment-resistant depression (TRD) patients under placebo [37].

#### 2.1.5. Lateral Habenula and Antidepressant Effects

Yang et al. demonstrated that ketamine-induced blocking of NMDAR-dependent neuronal bursting in the lateral habenula (LHb) disinhibits downstream monoaminergic reward centers [38]. The LHb is considered an “anti-reward” center, and its hyperactivity is associated with depressive phenotypes [39,40,41]. Ketamine reverses this hyperactivity by blocking NMDA receptors and low-voltage-sensitive T-type calcium channels (T-VSCCs) in the LHb, inducing rapid antidepressant effects [38,39,40]. However, this effect is temporary, ceasing as ketamine is cleared from the body [40].

#### 2.1.6. Salience Network and Default Mode Network

Marguilho, Figueiredo and Castro-Rodrigues propose a two-dose ketamine model grounded in neuroscience and phenomenology, integrating insights from the REBUS model and predictive self-binding theory [41]. Their approach distinguishes between two key networks involved in self-experience: the Salience Network (SN), associated with bodily or “embodied” self-consciousness, and the Default Mode Network (DMN), linked to the narrative or autobiographical self.

They suggest that: low doses (0.25–0.5 mg/kg) primarily affect the SN, producing dissociative, empathogenic experiences by loosening bodily self-awareness, especially when contextual influence is minimal; moderate doses (0.5–1 mg/kg) more strongly impact the DMN, inducing psychedelic experiences by disrupting narrative self-models, particularly in supportive therapeutic settings [41].

The model posits that these differential effects align with ketamine’s unique pharmacology—especially its NMDA receptor distribution—and are supported by psychometric and phenomenological data, as well as historical clinical use. This framework serves as the theoretical foundation for their proposed protocol using two distinct ketamine sessions, each targeting different dimensions of self-processing, to optimize therapeutic outcomes in psychedelic-assisted psychotherapy [41].

### 2.2. Short-Course Ketamine Protocol

Patients who struggle to cope with the diagnosis or whose care is focused on disease control rather than remission (“palliative care”) require interventions that are both effective and prompt [42]. These interventions must not add further suffering—that is, they should avoid side effects that may compromise quality of life or functional status [43]. The decision to limit the number of sessions in a model designed for palliative care might be based on favorable evidence from similar protocols in areas such as substance use disorders, mood disorders, and pain.

#### 2.2.1. Ketamine and Substance Use Disorders

In the context of substance use disorder (SUD), ketamine has been studied for its potential benefits in individuals with alcohol, cocaine, and opioid dependence [44].

In a randomized controlled trial involving individuals with alcohol use disorder [45], three sessions of ketamine (0.8 mg/kg) administered over the course of one week, in combination with psychotherapy or alcohol education, led to a greater number of sustained abstinent days compared to the placebo group.

For cocaine dependence, another randomized controlled trial [46] demonstrated that five consecutive days of 40 min intravenous ketamine infusions, combined with mindfulness-based relapse prevention, resulted in a nearly fivefold increase in abstinent individuals after two weeks compared to placebo. Furthermore, participants in the ketamine group had a 53% lower likelihood of relapses and reported a 58.1% reduction in craving.

In patients with opioid dependence, ketamine has shown promising results, with evidence suggesting it can alleviate acute withdrawal symptoms and prolong abstinence. Although repeated administrations have proven more effective for achieving desired outcomes, even studies with a limited number of sessions have reported meaningful effects [47]. It is worth noting that, among the oncologic population targeted by the model proposed in this study, opioid use is common for the management of pain and other comorbid conditions [48]. In patients with cancer-related pain, the prevalence of substance dependence is approximately 8%, while harmful use is observed in around 23.5% [49]. Therefore, the potential of a medication such as ketamine to replace or at least reduce opioid consumption is of particular interest.

#### 2.2.2. Ketamine and Mood

The rapid antidepressant effects of ketamine have been observed in patients with treatment-resistant depression. In one study [50], 24 h after a single intravenous infusion, 27% of participants met response criteria and 5% achieved remission. Responders exhibited a mean reduction of 22.3 points in MADRS scores. No antidepressant response was observed at any time point in the active control group. In a more robust demonstration of early efficacy, a large randomized controlled trial assessing intranasal esketamine [51] reported remission rates of 20.4% in the esketamine group compared to 9.8% in the placebo group at 24 h after the first dose. Response rates were also higher in the esketamine group (34.5%) versus placebo (25.3%). Rapid antidepressant effect is particularly relevant in acute scenarios where a rapid reduction in depressive symptoms is critical—such as during a suicidal crisis, while awaiting the delayed onset of conventional antidepressants or even in palliative care.

In a recent RCT [52], following a single infusion, the most prominent improvement was observed in depressive symptoms specifically related to sadness. Notably, symptoms such as sadness and lassitude demonstrated sustained improvement during the first week after the intervention.

To evaluate the time to relapse following a single ketamine infusion in patients with treatment-resistant depression (TRD) during the follow-up phase (days 3 to 30), a randomized controlled trial (RCT) was conducted [53]. The study demonstrated that both remission and response rates were dose-dependent—the higher the dose, the greater the rates of both outcomes. Sustained remission and response over time also followed a dose-dependent pattern. Among remitters, 74%, 47%, and 26% remained in remission at days 7, 14, and 30, respectively. Among responders, 81%, 52%, and 33% maintained their response at the same time points.

Suicidal ideation has been shown to decrease following a single dose of ketamine, with effects emerging within the first 72 h post-administration [54]. This phenomenon has been more extensively studied with intravenous racemic ketamine and warrants further investigation across alternative formulations.

Regarding cancer patients, an open-label phase 2 study [55] showed that three intranasal ketamine sessions given over the course of one week led to a reduction in depression severity after 8 days. Notably, this improvement was partially maintained 14 days after the final dose, without any additional administration. A narrative review of studies using ketamine-assisted psychotherapy (KAP) as a primary intervention to address psycho-existential distress in patients with severe medical conditions [16] found that the most common approach involved a single intravenous or intramuscular infusion. In most of these studies, depressive mood was the primary outcome, and significant improvement was reported after the intervention.

#### 2.2.3. Ketamine and Pain

Orhurhu and colleagues [55] analyzed seven randomized controlled trials (n = 211 patients) investigating the use of ketamine for chronic pain. In most studies, pain scores improved and remained reduced for up to two weeks after a single session. On average, the studies involved one ketamine session, with durations ranging from a few hours to up to ten days [56].

Regarding acute pain and hyperalgesia, a meta-analysis by Thompson and colleagues [57] examined controlled trials employing experimental models of acute pain and hyperalgesia to assess the analgesic effects of NMDA receptor antagonists. Among 70 studies included in the analysis, 54 specifically evaluated ketamine and its derivatives. Across all studies, a single low-dose ketamine infusion produced a robust analgesic effect, reducing pain scores by an average of 26%.

As for cancer-related pain, a review of 35 randomized controlled trials [58] demonstrated that ketamine can significantly reduce pain intensity in oncology patients. Again, even with a limited number of infusions, analgesic effects were observed for up to two weeks post-administration compared to control conditions.

Thus, the use of ketamine in oncology patients—whether palliative or not—emerges as a promising strategy to expand the therapeutic arsenal available for this population

### 2.3. Ketamine Models of Care

Currently, three primary models of ketamine used in mental health care have been identified: the biochemical model, the psychotherapeutic model, and the psychedelic model (Table 1) [59].

In the biochemical model, the primary focus is on ketamine’s biological (neurochemical) effects, with an emphasis on minimizing its dissociative and psychedelic properties—which are often considered undesirable side effects [60,61]. This approach pays little attention to the patient’s mental state or the “set and setting” during treatment. Patients are typically viewed as passive recipients of medication, undergoing six or more ketamine infusions over two to three weeks, with sessions spaced closely together to produce a cumulative effect. The standard protocol involves intravenous administration at 0.5 mg/kg over 40 min [62]. This model is particularly suitable for individuals with complex medical comorbidities (e.g., cardiovascular disease) who require a more controlled and predictable pharmacological intervention [18]. However, one of its main criticisms is the lack of psychological preparation and integration—essential components often overlooked by providers outside the mental health field [59,63].

The psychotherapeutic model integrates ketamine as an adjunct to psychotherapy, using it to enhance verbal expression, emotional processing, and insight during sessions [59,64]. Rather than focusing solely on symptom relief, this approach emphasizes self-exploration and behavioral change within a therapeutic alliance. Patients receive sub-psychedelic doses (0.25–1.0 mg/kg), which can be administered intramuscularly, subcutaneously, via nasal spray, or as sublingual lozenges—depending on local availability [59,60]. During treatment, individuals revisit personally significant themes, gaining insights and consolidating therapeutic progress. To maximize sustained benefits, ketamine sessions are typically spaced 3–6 weeks apart, with regular talk-therapy in between to help integrate psychological material emerging from each ketamine experience [39,59,64].

In the psychedelic model, ketamine is used explicitly for its dissociative and psychedelic effects, inducing a temporary altered state of consciousness often characterized by dreamlike visions and deep introspective experiences [59,65]. Because these states can be powerful, specialized psychological support is essential before, during, and after administration. This model often incorporates ritualistic or ceremonial elements—music, eye coverings, guided imagery—to create a safe, supportive therapeutic container [7]. Ketamine is typically administered via intramuscular injection at doses of 1.0–1.5 mg/kg (sometimes up to 2.0 mg/kg). Doses below 1.0 mg/kg are generally considered safe for administration in non-hospital settings without continuous monitoring [60,65]. However, this approach requires that individuals have sufficient psychological resilience—often cultivated through prior meditation practice or psychotherapy—to navigate intense altered states. A strong therapeutic alliance is paramount, since higher doses do not necessarily equate to better outcomes. Instead, proper preparation, integration, and psychological readiness are key to ensuring that the psychedelic experience leads to meaningful, lasting transformation [16,59,65].

In palliative care, ketamine has emerged as a promising intervention capable of addressing the multidimensional suffering experienced by patients with advanced illness. Evidence shows that, at sub-anesthetic doses, ketamine provides rapid and clinically meaningful relief across three critical domains: physical pain, psychological symptoms, and existential distress [16,66,67,68]. Its analgesic effects reduce both cancer-related and neuropathic pain while decreasing opioid requirements; its antidepressant and anxiolytic properties alleviate mood disturbances and anxiety often resistant to conventional treatments; and its capacity to relieve demoralization and death-related anxiety contributes to improved spiritual well-being and quality of life at the end of life. Although these effects are typically time-limited, they highlight ketamine’s unique therapeutic profile as a multimodal agent that simultaneously mitigates physical, emotional, and existential suffering in palliative care populations.”

While the FDA breakthrough therapy designation of intranasal esketamine (Spravato^®^) underscores the growing recognition of ketamine’s antidepressant potential, its approval reflects a predominantly pharmacological model aimed at symptom reduction in treatment-resistant depression. In contrast, KAP emphasizes an integrative approach in which the psychoactive experience is embedded within structured therapeutic preparation and integration, targeting not only mood symptoms but also the existential, spiritual, and relational dimensions of suffering that are particularly salient in palliative care. This contrast highlights the need to differentiate between ketamine as a stand-alone drug therapy and KAP as a multimodal intervention with broader implications for meaning-making and quality of life at the end of life.

## 3. Results

After reviewing the foundational rationale, the subsequent sections present a short-course ketamine-assisted psychotherapy model tailored to palliative care populations. The model integrates preparatory and integrative psychotherapy, two dosing sessions (low and moderate dose), goals of care discussion (GOCD), concurrent psychotherapy, and optional pharmacological optimization. Drawing from both biochemical and psychedelic paradigms, this model offers a compassionate, flexible, and patient-centered approach to relieving suffering and enhancing meaning at the end of life. Figure 1 illustrates the neural pathways implicated in the dissociative–psychedelic model of ketamine.

Table 2 outlines the essential therapeutic elements that define the model’s structure and clinical approach. These components reflect an integration of paradigms tailored to the unique psychological and existential needs of individuals receiving palliative care. Figure 1 provides a visual representation of the proposed treatment sequence, illustrating the temporal relationship between preparatory, dosing, and integration phases across the two ketamine sessions. Together, the table and figure offer a comprehensive overview of both the content and delivery of this clinical framework. The main features of the proposed model are presented below.

### 3.1. Psychotherapy-Grade Therapeutic Frame

A consistent therapeutic frame is essential for any psychotherapeutic intervention, including those involving ketamine. The proposed model adopts a psychotherapy-grade frame, characterized by:A clear, time-limited treatment plan.Shared understanding of goals and expectations.Boundaries for communication and follow-up.Proactive exploration of treatment termination.

Particularly in palliative settings, discussing the end of treatment from the outset helps prevent regression, ambivalence, and existential despair. The time-limited nature of the protocol (two ketamine sessions plus preparatory/integration sessions) reflects both clinical pragmatism and therapeutic intent.

### 3.2. Treatment Phases and Timeline

#### 3.2.1. Preparation Phase (2–3 Sessions)

Prior to ketamine administration, patients engage in structured preparation with a trained clinician. This phase involves:Psychoeducation about ketamine’s mechanisms, limitations, and potential benefits.Exploration and moderation of treatment expectations.Establishment of at least three goals of care tailored to the patient’s values and capacity.Orientation to potential psychological content or altered states during ketamine sessions.

These preparatory conversations also emphasize that ketamine is not a panacea, but rather a tool within a broader, active process of adaptation and meaning-making.

#### 3.2.2. Ketamine Sessions (2 Sessions)

To enhance the feasibility of the model, we opted for two intravenous ketamine sessions, each administered as a continuous 40 min infusion, with at least one day between them. It is important to note that this protocol is supposed to be applied to inpatients in specialized high-complexity oncology hospitals. Therefore, a protocol involving multiple sessions or requiring patients to return for additional visits would not be practical in this setting. It should also be considered that many of the patients included in the proposed protocol should be in palliative care. In such cases, each moment becomes crucial when planning interventions. Rapid-onset treatments take precedence over those that require longer periods or multiple doses to yield the desired outcome.

Two ketamine dosing sessions are scheduled approximately one week apart:Session 1 (low-dose): ~0.3–0.5 mg/kg IV, primarily for acclimatization and to trigger dissociative experiences.Session 2 (moderate-dose): ~0.75–1.0 mg/kg IV, oriented toward deeper emotional or existential processing provided by psychedelic experiences.

Each session is conducted in a supportive, therapeutic setting, with a clinician present for safety and non-directive support. Non-verbal guidance, music, and comfort-enhancing measures may be used to deepen the experience.

#### 3.2.3. Integration Phase (2–3 Sessions)

Post-treatment integration sessions are conducted within 24–72 h of each ketamine experience and continue beyond the final dose. These sessions serve to:Process psychological insights or unresolved emotional material.Revisit and refine GOCD.Translate subjective experiences into actionable, values-based changes.Address potential “termination reactions” and create closure.

In patients nearing the end of life, this process may support existential resolution, legacy work, or emotionally significant conversations.

### 3.3. Concomitant Psychotherapy

Patients are encouraged (and ideally required) to begin weekly evidence-based psychotherapy at least two weeks prior to the first ketamine session. Acceptable modalities include Cognitive Behavioral Therapy (CBT), Meaning-Centered Psychotherapy, Dignity Therapy, Existential and Narrative Therapies, Mindfulness-based interventions, or other validated approaches suited to the individual’s needs.

Unlike traditional psychedelic therapy models, psychotherapy in this model is concurrent but not necessarily co-administered with ketamine. Therapists may be existing providers, community clinicians, or professionals integrated into the palliative care team. This approach is flexible and emphasizes strong therapeutic alliances over specific techniques.

Benefits of this structure include:Enhanced engagement due to ketamine’s rapid cognitive/emotional effects.Psychotherapeutic containment of challenging content.A continuity pathway for psychological support beyond the ketamine course.

Selecting appropriate outcome measures is critical for evaluating the impact of KAP in palliative care populations, where psychological, existential, and spiritual dimensions of suffering are often intertwined. Evidence from related interventions, such as Dignity Therapy, highlights a range of validated instruments that could be integrated into KAP studies [69]. Recommended measures include multi-dimensional scales that capture both subjective experience and functional impact, such as the Patient Dignity Inventory (PDI) for dignity-related distress, the FACIT-Sp Meaning/Peace subscale for spiritual well-being, and the Distress Scale II (DS-II) or Edmonton Symptom Assessment System—spiritual domain (ESAS-Sp) for existential distress [69,70]. Short-term assessments immediately post-session can capture acute ketamine effects, while longitudinal follow-ups (1–4 weeks) are essential to evaluate sustained outcomes. Complementary qualitative approaches, including semi-structured interviews, provide nuanced insight into meaning-making, transcendence, and spiritual experiences that may not be fully captured by standardized scales. Integrating these quantitative and qualitative measures offers a comprehensive framework to assess both the psychological and existential efficacy of KAP in palliative care.

### 3.4. Goals of Care

Drawing from GOCD principles, patients are supported in creating and integrating their goals into care planning, such as:Reconnecting with loved ones.Engaging in spiritually or personally meaningful activities.Establishing comfort-enhancing routines or rituals.Exploring creative expression or legacy documentation.

These goals are designed to be attainable within the patient’s health status, and the process promotes agency, coherence, and engagement even in the face of decline.

### 3.5. Bridging Biomedical and Psychedelic Paradigms

Rather than adopting a strictly biomedical or psychedelic approach, this model views ketamine-induced altered states as experiential opportunities—not essential for benefit, but often valuable for insight and emotional resolution. This orientation invites therapeutic meaning-making while maintaining clinical pragmatism.

The use of psychedelic-inspired session elements (e.g., music, eyeshades, supportive silence) enhances introspective depth without imposing a rigid metaphysical framework. Patients are encouraged to explore the meaning of their experiences within their own belief systems.

### 3.6. Clinical and Ethical Considerations

KAP in palliative care raises important ethical considerations that must be addressed to ensure patient-centered and responsible practice. These include respecting patient autonomy and informed consent, particularly given the altered states of consciousness induced by ketamine and carefully weighing potential benefits against risks in a population often characterized by vulnerability and frailty. Ethical implementation also requires transparency regarding therapeutic goals, the limits of evidence, and potential side effects, as well as sensitivity to cultural and spiritual dimensions of end-of-life care. Incorporating ongoing ethical reflection, multidisciplinary consultation, and robust documentation can help safeguard patients’ dignity, promote trust, and support equitable access to KAP within palliative care frameworks.

This model is designed to be:Time-limited and feasible within palliative care settings.Flexible, allowing for individualized psychotherapy and diverse treatment goals.Ethically grounded, emphasizing autonomy, emotional safety, and informed consent.Integrated, with communication between ketamine providers, psychotherapists, and palliative care teams.

Special attention is given to psychological containment, realistic expectation-setting, and post-treatment continuity of care (Figure 2).

### 3.7. Contraindications, Adverse Events and Monitoring Requirements

Ketamine, when administered at subanesthetic doses in palliative care, is generally well tolerated, but several adverse effects may occur. Short-term effects include dissociation, psychotomimetic symptoms, transient anxiety, blurred vision, dizziness, headache, nausea, and vomiting [71]. Cardiovascular effects are common, with temporary increases in systolic and diastolic blood pressure (typically 8–20 mmHg and 8–13 mmHg, respectively) and heart rate, peaking within 30–60 min of infusion and resolving within 1–2 h. Less frequent effects include chest discomfort or palpitations [72]. Patients with psychotic features, uncontrolled hypertension, or significant cardiovascular disease are generally considered contraindicated due to heightened risk [71,72].

Longer-term adverse effects, such as abuse potential, neurotoxicity, bladder dysfunction, and hepatotoxicity, have primarily been observed in the context of recreational or chronic misuse and have not been reported in therapeutic settings [73]. Nevertheless, careful patient selection, pre-infusion screening for psychiatric, cardiovascular, and substance use conditions, individualized dosing, and structured follow-up remain essential. Monitoring requirements include continuous observation of cardiovascular parameters during and shortly after infusion, readiness to manage acute hemodynamic changes, and structured assessment of psychological responses to optimize safety and therapeutic outcomes in palliative care populations [72,73].

## 4. Discussion

Patients in palliative care often face forms of suffering that go beyond the physical and biomedical domains, including existential distress, anticipatory grief, loss of meaning, and fear of death [16,74,75]. These deeply human and subjective aspects are not always adequately addressed by conventional approaches to care. In this context, the (re)emergence of ketamine as a psychotherapeutic agent presents an unprecedented opportunity to expand the scope of palliative medicine—particularly when combined with structured psychological interventions [16]. In this article, we propose an intensive, integrative model of ketamine-assisted psychotherapy (KAP) specifically designed for this population, structured around two ketamine experiences—a lower-dose dissociative session and a higher-dose psychedelic session—each framed by preparatory and integrative psychotherapy.

This model draws from emerging clinical practice and the growing body of literature on ketamine and psychedelic-assisted therapy, adapting core therapeutic principles to the sensitive and urgent context of end-of-life care [16,76,77]. Rather than privileging repeated pharmacological interventions, our approach emphasizes the depth of experience and the therapeutic relationship as central vectors of change. The initial dissociative-dose session is intended to introduce the patient safely to ketamine’s altered state of consciousness, reduce anticipatory anxiety, and strengthen the therapeutic alliance [16,77]. The subsequent psychedelic-dose session provides space for more intense and potentially transformative experiences, such as transcendence, biographical reconciliation, or ego dissolution—phenomena increasingly described in the literature as sources of profound psychological relief [16,76].

The phased structure—preparation, dissociative experience, psychedelic experience, and integration—creates a therapeutic rhythm that respects patients’ emotional, cognitive, and somatic limits while leveraging windows of psychological plasticity and openness [16]. This approach aligns with the view that, in the context of existential suffering, shifts in meaning and perspective may be more therapeutically relevant than conventional symptom reduction [75,76]. More than simply reducing depression or anxiety, this model seeks to facilitate re-signification of the dying process, expansion of temporal perspective, reconnection with personal values, and acceptance of impermanence—goals deeply consistent with the ethos of palliative medicine [75].

Clinical experience in palliative care has shown that patients often seek not only symptom relief but the possibility of finding peace, reconciliation, and meaning before death [75,77]. Brief yet intensive psychotherapeutic models, such as the one proposed here, may offer a realistic and ethically sound way to meet these needs, even when time is short and resources are limited. The potential to evoke experiences of reconnection—with loved ones, with one’s life story, or with spiritual dimensions—can yield profound therapeutic value for those facing the end of life [76,77].

The proposed intensive model of ketamine-assisted psychotherapy (KAP) for palliative care could be compared to the Montreal model for treatment-resistant depression (TRD) [78], while adapting key elements to meet the unique clinical, emotional, and existential demands of patients nearing the end of life. Both approaches share an integrative ethos that bridges biomedical and psychedelic paradigms, recognizing ketamine not only as a pharmacological agent but also as a psychological catalyst. However, significant differences emerge in terms of treatment goals, structure, dosing, and clinical context.

The Montreal model was developed through years of real-world psychiatric practice and research, with the aim of offering a scalable, biopsychosocial framework for the treatment of severe, chronic depression [78]. It typically involves six ketamine sessions spaced over several weeks, interwoven with preparation, integration, and adjunctive psychotherapy. The model emphasizes psychological framing, experiential learning, and a rolling therapeutic rhythm that supports incremental change and adaptation. Ketamine is generally administered in moderate dissociative doses, sometimes edging toward psychedelic intensity, with a focus on maintaining clinical safety and tolerability across a broad range of psychiatric presentations.

In contrast, the palliative care model is designed for brevity, depth, and existential relevance. Rather than a rolling, multi-session course, it consists of two core ketamine experiences: an initial low-dose (dissociative) session followed by a higher-dose (psychedelic) session, each embedded within dedicated preparatory and integrative psychotherapy. This condensed structure acknowledges the limited time horizons and emotional resources of many palliative care patients, while seeking to maximize therapeutic impact through meaningful, well-supported altered-state experiences. The dissociative session serves to familiarize the patient with ketamine’s effects, reduce anxiety, and strengthen trust in the therapeutic process, whereas the psychedelic session is intended to offer the possibility of symbolic resolution, transcendence, or biographical reconciliation [16,77].

The two models also diverge in their therapeutic aims. While the Montreal model focuses on symptom reduction, behavioral change, and psychological insight in the context of chronic depression, the palliative care model prioritizes existential healing—helping patients find peace, meaning, and acceptance as they approach death [16]. In this way, the use of ketamine in palliative settings is less concerned with long-term remission and more aligned with goals central to palliative medicine: dignity, emotional relief, relational closure, and spiritual integration [75].

From a pragmatic standpoint, ketamine’s short duration of action, high tolerability, and relative flexibility make it particularly well-suited for both models, albeit for different reasons [16]. In the Montreal model, its short half-life facilitates outpatient use and repeated exposure; in the palliative care model, it allows for intensive but time-sensitive interventions that respect patients’ physical and emotional limitations [1,4]. Moreover, ketamine’s capacity to catalyze non-ordinary states of consciousness without the prolonged effects associated with classic psychedelics (e.g., psilocybin, LSD) makes it a uniquely viable candidate for use in medical and end-of-life contexts where control, safety, and adaptability are paramount [76].

While both the Montreal model and the palliative model share a commitment to the therapeutic use of ketamine within structured, psychologically informed settings, they reflect distinct clinical logics. The former is geared toward chronicity, complexity, and long-term psychiatric care; the latter toward immediacy, meaning-making, and existential closure. Their complementarity underscores the versatility of ketamine as a therapeutic tool and highlights the importance of tailoring psychedelic-assisted psychotherapy to the particular needs, vulnerabilities, and timeframes of diverse patient populations [16,78].

Similarly, psilocybin-assisted psychotherapy (PAP) protocols share core principles of safety, preparation, and integration, while varying in dose, format, and therapeutic focus. Therapeutic doses for depression typically center around 25 mg, with some studies using weight-adjusted or preparatory lower-dose sessions, which may themselves have antidepressant effects [5]. Most protocols employ individualized therapy, emphasizing exploration of depressive thoughts and behaviors, whereas in long-term illness populations, therapy may include group sessions and focus on illness-related distress, relational support, and existential meaning [79]. Across all models, psychotherapy is essential not only for safety during dosing but for guiding insight, reframing symptoms, and supporting long-term psychological and relational growth.

Implementing KAP in palliative care settings presents several challenges, including managing patient vulnerability, potential hemodynamic instability, and the integration of profound psychological experiences into ongoing care. Additional considerations involve variability in patient responsiveness, risk of dissociative or distressing experiences, and the need for coordination with multidisciplinary teams. To mitigate these risks, structured screening protocols, individualized dosing regimens, continuous physiological monitoring, and thorough preparatory and integrative sessions are essential. Furthermore, training clinicians in trauma-informed and psychedelic-informed care, alongside establishing clear emergency procedures and ethical safeguards, can enhance both safety and therapeutic efficacy while ensuring that patients’ autonomy and comfort remain central throughout the KAP process.

Despite its promise, the integration of KAP into palliative care and public health systems faces substantial challenges, including high drug costs, regulatory hurdles, infrastructure demands, and the need for systematic training of multidisciplinary teams. Beyond technical expertise, therapists must cultivate empathetic presence, trust-building, spiritual intelligence, and ethical integrity to guide patients through altered states of consciousness and support the integration of insights into coping and existential frameworks [80]. Demonstrating cost-effectiveness and real-world feasibility will be essential for policymakers, but equally critical is the recognition that therapist preparation extends beyond pharmacological knowledge to encompass experiential and relational competencies that ensure KAP can be delivered safely, ethically, and meaningfully to populations with high unmet needs.

That said, several ethical and practical challenges must be acknowledged. Careful assessment of decision-making capacity, management of expectations, the presence of skilled psychotherapeutic support, and a strong emphasis on patient-centered care are essential to ensuring that ketamine is used safely and ethically in this population [75,77]. The involvement of family members and the broader care team can further support the therapeutic process, helping to extend the benefits of the intervention beyond the individual to the relational and social context surrounding the patient [75].

Ongoing clinical trials are beginning to explore the feasibility and impact of ketamine-based interventions in palliative care. Current studies range from intranasal administration for depression and anxiety in early-stage cancer patients [81], to KAP for terminally ill individuals facing death-related distress [81], as well as trials assessing subcutaneous, oral, and combined approaches for pain and mood symptoms in advanced illness [82]. While still preliminary, these investigations emphasize feasibility, safety, and integration of caregivers, and they hold promise for expanding therapeutic options in addressing the complex psychological, existential, and physical dimensions of suffering at the end of life.

It is important to highlight that the rationale for a two-session KAP model in palliative care draws from both empirical findings in adjacent populations and the unique temporal constraints of terminal illness. While single-dose ketamine has demonstrated rapid antidepressant effects in treatment-resistant depression [38,44,47,51,54,64,65], individuals facing life-limiting conditions often require an approach that not only alleviates mood symptoms but also addresses existential, spiritual, and relational dimensions of suffering. In this context, a time-limited, two-session format represents a pragmatic compromise: intensive enough to facilitate meaningful psychotherapeutic work within the short prognosis typical of palliative care, yet sufficiently streamlined to remain feasible in clinical practice. Although direct evidence in palliative populations is currently lacking, this model represents a theoretically grounded extrapolation, one that prioritizes patient needs for transcendence, legacy work, and existential resolution in the limited timeframe available. Future qualitative and case-based studies will be critical to further substantiate and refine this approach.

Future research should also assess the clinical efficacy, emotional safety, and existential impact of this model across diverse palliative care populations, using both quantitative and qualitative outcome measures that go beyond traditional symptom metrics [16,76]. It will also be important to explore how psychosocial, spiritual, and relational factors—such as prior trauma, spiritual beliefs, and interpersonal dynamics—influence treatment response. Logistical and systemic considerations such as staff training, infrastructure requirements, and cost-effectiveness must also be addressed to facilitate real-world implementation [16].

In sum, KAP—when applied with sensitivity, structure, and collaboration—may represent a meaningful innovation in the care of patients approaching the end of life. This brief but intensive two-session model, supported by robust psychotherapy, aims not only to alleviate suffering but to restore dignity, foster reconnection, and offer meaning at one of the most vulnerable and significant junctures of human experience [16,75,77].

## 5. Conclusions

Unlike classical psychedelics such as psilocybin or LSD, ketamine offers practical advantages for use in palliative contexts: a short duration of action, a well-established safety profile even in medically fragile populations, and flexibility in terms of clinical setting. Ketamine’s responsiveness to therapeutic context (or set and setting) enables clinicians to tailor the experience to the patient’s specific needs, minimizing risk while maximizing potential benefit—even when physical or emotional fragility is present.

## Figures and Tables

**Figure 1 healthcare-13-02714-f001:**
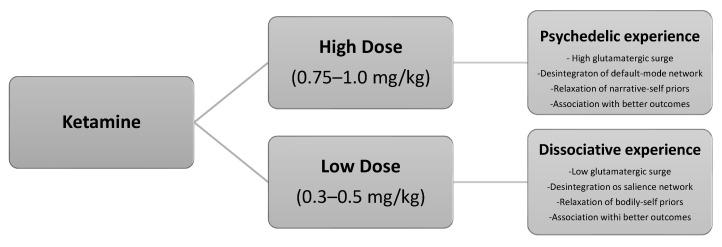
The dissociative–psychedelic model of ketamine.

**Figure 2 healthcare-13-02714-f002:**
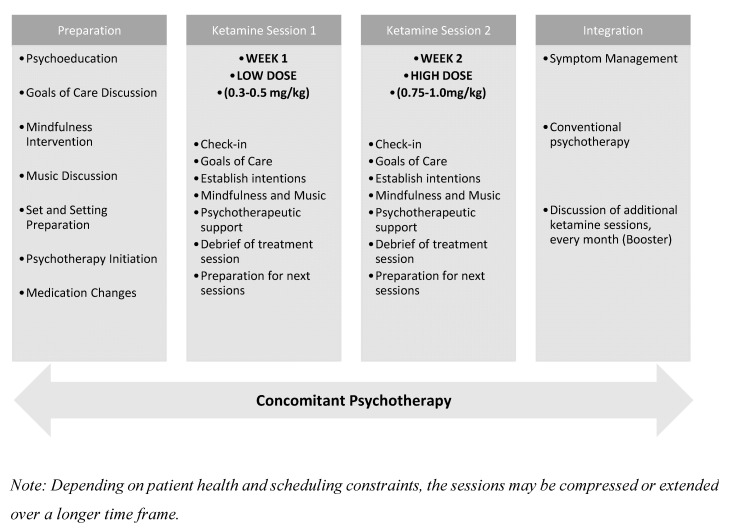
Timeline of the Two-Session Ketamine-Assisted Psychotherapy Model.

**Table 1 healthcare-13-02714-t001:** Ketamine Clinic Treatment Models.

Model	Biochemical	Psychotherapeutic	Psychedelic
Name	Ketamine Infusion Therapy	Ketamine-Assisted Psychotherapy	Psychedelic Therapy
Objective	Symptom management	Catalyzing psychological change	Facilitating transformative experiences
Focus	Medication effects	Psychotherapy	Subjective experience
Number of Sessions	6 to 12 sessions	3 to 4 sessions	1 to 2 sessions
Procedure	No preparation or integration	Includes preparation and integration	May or may not include preparation and integration
Administration Routes	Intravenous	Oral/Sublingual/Intramuscular/Subcutaneous	Intramuscular/Subcutaneous
Treatment Setting	Procedure room	Therapy office	Therapy offices & group settings

**Table 2 healthcare-13-02714-t002:** Core Components of the Proposed Ketamine-Assisted Psychotherapy Model for Palliative Care.

Component	Description
Population	Patients receiving palliative care; capable of engaging in brief psychological interventions
Dose & Administration	Two intravenous ketamine sessions (0.3–1.0 mg/kg over ~40 min), spaced ~5–10 days apart
Psychological Framework	Preparation, experience, and integration; emphasis on meaning-making, emotional exploration, and patient-centered care
Setting	Quiet, comfortable, non-medicalized room; minimal clinical cues
Sensory Modulation	Use of blindfolds and curated music playlists to enhance immersive and symbolic experience
Framing of Experience	Experiences presented as potentially meaningful reflections of inner states, not random or pathological
Therapeutic Stance	Emotional validation over interpretation; process emphasized over content
Patient Involvement	Collaborative selection of music, language, and therapeutic orientation; flexibility in session goals
Integration	Post-session reflection emphasizing emotional insights, symbolic coherence, and existential relevance

## Data Availability

No new data were created or analyzed in this study.

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
