# Peer review of "Unfolding States of Mind: A Dissociative-Psychedelic Model of Ketamine-Assisted Psychotherapy in Palliative Care"

_healthcare, 2025, doi:10.3390/healthcare13212714_

Round 1

Reviewer 1 Report

Comments and Suggestions for Authors

The review is well-articulated with a balanced perspective. The topic is innovative and likely to capture the reader’s interest. In the Materials and Methods section, consider including a subsection on the effects of ketamine on brain chemistry, particularly its influence on neurotransmitters.

Author Response

The review is well-articulated with a balanced perspective. The topic is innovative and likely to capture the reader’s interest. In the Materials and Methods section, consider including a subsection on the effects of ketamine on brain chemistry, particularly its influence on neurotransmitters.

R: To expand on the effects of ketamine on brain chemistry, particularly its influence on neurotransmitters, the following paragraphs were included and subsection “2.1.1 NMDA Antagonism” had its name changed to “2.1.1 NMDA Antagonism and Neurotransmitter Influence”:

“In addition to its glutamatergic actions, ketamine modulates other neurotransmitter systems. It transiently increases extracellular serotonin, dopamine, and norepinephrine levels in the prefrontal cortex via glutamatergic projections to monoaminergic nuclei, which may contribute to improvements in mood and, notably, anhedonia. Evidence also indicates that ketamine upregulates GABAA receptor activity and enhances GABA turnover, although excessive GABAergic potentiation (e.g., from benzodiazepines) appears to blunt its antidepressant efficacy.

Finally, ketamine interacts with the opioid system, showing moderate affinity for mu- and kappa-opioid receptors. Clinical studies have reported that opioid receptor antagonism can abolish ketamine’s antidepressant response, suggesting a modulatory role of opioid signaling. Taken together, ketamine’s antidepressant profile emerges from the convergence of glutamatergic disinhibition, transient monoaminergic enhancement, GABAergic modulation, and opioid receptor interactions, distinguishing it mechanistically from conventional monoaminergic antidepressants.”

Reviewer 2 Report

Comments and Suggestions for Authors

The article entitled "Unfolding States of Mind: A Dissociative-Psychedelic Model of Ketamine-Assisted Psychotherapy in Palliative Care" is overall interesting, timely, and provides a good overview of recent studies in the field of ketamine-assisted psychotherapy for palliative care. However, it requires a major revision before being considered for acceptance. The following comments are suggested to improve the manuscript:

  1. Mechanism of Action Representation

    • Although the manuscript explains the possible mechanism of action of ketamine in text, it is not supported by any pictorial/diagrammatic representation.

    • A schematic figure or flowchart illustrating the dissociative-psychedelic model and molecular/neurological pathways of ketamine would greatly enhance reader comprehension.

  2. Clinical Relevance and Case Evidence

    • The manuscript would benefit from the inclusion of specific case examples or clinical data demonstrating how ketamine-assisted psychotherapy has impacted palliative care patients (e.g., improvements in pain, anxiety, or existential distress). The breakthrough designation of spravato (esketamine) can be discussed.

  3. Challenges and side effects of ketamine

    • While ketamine is the focus, the inclusion of challenges and risk mitigation strategies of ketamine would add context and strengthen the discussion.

  4. Safety and Ethical Considerations

    • The manuscript should address safety concerns, ethical issues, regulatory frameworks, and guidelines of psychedelics, which were drafted recently.

    • A subsection on contraindications, adverse events, and monitoring requirements would balance the positive therapeutic outlook.

  5. Recent Literature Update

    • The review should incorporate the latest clinical trial updates (2022–2025), particularly from NCT-registered studies using clinicaltrial.gov and the European clinical trial registry, to ensure the discussion remains current.

Author Response

1-Mechanism of Action Representation

Although the manuscript explains the possible mechanism of action of ketamine in text, it is not supported by any pictorial/diagrammatic representation.

A schematic figure or flowchart illustrating the dissociative-psychedelic model and molecular/neurological pathways of ketamine would greatly enhance reader comprehension.

R: A schematic figure was included to illustrate the dissociative-psychedelic model (figure 1).

2-Clinical Relevance and Case Evidence

The manuscript would benefit from the inclusion of specific case examples or clinical data demonstrating how ketamine-assisted psychotherapy has impacted palliative care patients (e.g., improvements in pain, anxiety, or existential distress). The breakthrough designation of spravato (esketamine) can be discussed.

R: To adress the impact of ketamine for palliative care patients and the designation of Spravato the following paragraphs were added at the end of the Material and Methods section:

“In palliative care, ketamine has emerged as a promising intervention capable of addressing the multidimensional suffering experienced by patients with advanced illness. Evidence shows that, at sub-anesthetic doses, ketamine provides rapid and clinically meaningful relief across three critical domains: physical pain, psychological symptoms, and existential distress. Its analgesic effects reduce both cancer-related and neuropathic pain while decreasing opioid requirements; its antidepressant and anxiolytic properties alleviate mood disturbances and anxiety often resistant to conventional treatments; and its capacity to relieve demoralization and death-related anxiety contributes to improved spiritual well-being and quality of life at the end of life. Although these effects are typically time-limited, they highlight ketamine’s unique therapeutic profile as a multimodal agent that simultaneously mitigates physical, emotional, and existential suffering in palliative care populations.”

“While the FDA breakthrough therapy designation of intranasal esketamine (Spravato) underscores the growing recognition of ketamine’s antidepressant potential, its approval reflects a predominantly pharmacological model aimed at symptom reduction in treatment-resistant depression. In contrast, ketamine-assisted psychotherapy (KAP) emphasizes an integrative approach in which the psychoactive experience is embedded within structured therapeutic preparation and integration, targeting not only mood symptoms but also the existential, spiritual, and relational dimensions of suffering that are particularly salient in palliative care. This contrast highlights the need to differentiate between ketamine as a stand-alone drug therapy and KAP as a multimodal intervention with broader implications for meaning-making and quality of life at the end of life”.

3-Challenges and side effects of ketamine

While ketamine is the focus, the inclusion of challenges and risk mitigation strategies of ketamine would add context and strengthen the discussion.

R: To address this issue, the following paragraph was added to the Discussion Section:

“Implementing ketamine-assisted psychotherapy (KAP) in palliative care settings presents several challenges, including managing patient vulnerability, potential hemodynamic instability, and the integration of profound psychological experiences into ongoing care. Additional considerations involve variability in patient responsiveness, risk of dissociative or distressing experiences, and the need for coordination with multidisciplinary teams. To mitigate these risks, structured screening protocols, individualized dosing regimens, continuous physiological monitoring, and thorough preparatory and integrative sessions are essential. Furthermore, training clinicians in trauma-informed and psychedelic-informed care, alongside establishing clear emergency procedures and ethical safeguards, can enhance both safety and therapeutic efficacy while ensuring that patients’ autonomy and comfort remain central throughout the KAP process.”

4-Safety and Ethical Considerations

The manuscript should address safety concerns, ethical issues, regulatory frameworks, and guidelines of psychedelics, which were drafted recently.

A subsection on contraindications, adverse events, and monitoring requirements would balance the positive therapeutic outlook.

R: To address these issues, the following paragraphs were included in the Results section. A new subsection on contraindications, adverse events, and monitoring requirements was created.

“Ketamine-assisted psychotherapy (KAP) in palliative care raises important ethical considerations that must be addressed to ensure patient-centered and responsible practice. These include respecting patient autonomy and informed consent, particularly given the altered states of consciousness induced by ketamine, and carefully weighing potential benefits against risks in a population often characterized by vulnerability and frailty. Ethical implementation also requires transparency regarding therapeutic goals, the limits of evidence, and potential side effects, as well as sensitivity to cultural and spiritual dimensions of end-of-life care. Incorporating ongoing ethical reflection, multidisciplinary consultation, and robust documentation can help safeguard patients’ dignity, promote trust, and support equitable access to KAP within palliative care frameworks.”

“3.7. Contraindications, adverse events, and monitoring requirements

Ketamine, when administered at subanesthetic doses in palliative care, is generally well tolerated, but several adverse effects may occur. Short-term effects include dissociation, psychotomimetic symptoms, transient anxiety, blurred vision, dizziness, headache, nausea, and vomiting. Cardiovascular effects are common, with temporary increases in systolic and diastolic blood pressure (typically 8–20 mmHg and 8–13 mmHg, respectively) and heart rate, peaking within 30–60 minutes of infusion and resolving within 1–2 hours. Less frequent effects include chest discomfort or palpitations. Patients with psychotic features, uncontrolled hypertension, or significant cardiovascular disease are generally considered contraindicated due to heightened risk.

Longer-term adverse effects, such as abuse potential, neurotoxicity, bladder dysfunction, and hepatotoxicity, have primarily been observed in the context of recreational or chronic misuse and have not been reported in therapeutic settings. Nevertheless, careful patient selection, pre-infusion screening for psychiatric, cardiovascular, and substance use conditions, individualized dosing, and structured follow-up remain essential. Monitoring requirements include continuous observation of cardiovascular parameters during and shortly after infusion, readiness to manage acute hemodynamic changes, and structured assessment of psychological responses to optimize safety and therapeutic outcomes in palliative care populations”.

5-Recent Literature Update

The review should incorporate the latest clinical trial updates (2022–2025), particularly from NCT-registered studies using clinicaltrial.gov and the European clinical trial registry, to ensure the discussion remains current.

R: To address this issue, the following paragraph was included in the Discussion section:

“Ongoing clinical trials are beginning to explore the feasibility and impact of ketamine-based interventions in palliative care. Current studies range from intranasal administration for depression and anxiety in early-stage cancer patients, to ketamine-assisted psychotherapy for terminally ill individuals facing death-related distress, as well as trials assessing subcutaneous, oral, and combined approaches for pain and mood symptoms in advanced illness. While still preliminary, these investigations emphasize feasibility, safety, and integration of caregivers, and they hold promise for expanding therapeutic options in addressing the complex psychological, existential, and physical dimensions of suffering at the end of life.”

Reviewer 3 Report

Comments and Suggestions for Authors

General Comments

The manuscript offers a relevant and ambitious addition to the new field of psychedelic-assisted therapy in palliative care.  For patients with life-limiting illnesses, it suggests a theoretically supported and empirically useful two-session ketamine-assisted psychotherapy (KAP) approach.  The manuscript is simply written and well-structured, and the confluence of phenomenology, psychotherapy, and neuroscience is praiseworthy.  One of the writers' strengths is how well they portray ketamine as a neurobiological agent and a stimulus for psychotherapy.  To increase rigor and clinical applicability, some areas need to be clarified, given more empirical support, and refined.

Comments to the authors

  1. There is no clinical evidence to support the use of a two-session KAP model in palliative populations, despite the well-established neurobiological justification.  The writers ought to make a clearer distinction between empirical data and theoretical extrapolation.  Qualitative or case-based reporting, for instance, could substantiate assertions on transcendence, legacy labor, or existential resolve.
  2. Instead of being a structured assessment, the "Materials and Methods" section mostly reads like a narrative evaluation.  Indicate clearly that this is a narrative/conceptual review or provide clarification on if systematic procedures (databases searched, inclusion criteria) were applied.
  3. The suggested model and the Montreal model for TRD are compared in the discussion.  This is useful, but to better contextualize the contribution, the comparison might be extended to other psychedelic protocols (such as psilocybin-assisted therapy in end-of-life care).
  4. The implementation of this strategy in oncology and palliative care settings presents logistical, regulatory, and resource problems that should be more clearly addressed in the research (e.g., staff training, legal access to ketamine, cost, cultural acceptance).
  5. Although existential healing is emphasized as a crucial result, the article offers no precise metrics for success.  Add validated instruments for existential distress, meaning, or dignity (e.g., Patient Dignity Inventory, FACIT-Sp).

Author Response

1-There is no clinical evidence to support the use of a two-session KAP model in palliative populations, despite the well-established neurobiological justification.  The writers ought to make a clearer distinction between empirical data and theoretical extrapolation.  Qualitative or case-based reporting, for instance, could substantiate assertions on transcendence, legacy labor, or existential resolve.

R: To address this issue, the following paragraph was added in the Discussion section:

“The rationale for a two-session KAP model in palliative care draws from both empirical findings in adjacent populations and the unique temporal constraints of terminal illness. While single-dose ketamine has demonstrated rapid antidepressant effects in treatment-resistant depression, individuals facing life-limiting conditions often require an approach that not only alleviates mood symptoms but also addresses existential, spiritual, and relational dimensions of suffering. In this context, a time-limited, two-session format represents a pragmatic compromise: intensive enough to facilitate meaningful psychotherapeutic work within the short prognosis typical of palliative care, yet sufficiently streamlined to remain feasible in clinical practice. Although direct evidence in palliative populations is currently lacking, this model represents a theoretically grounded extrapolation, one that prioritizes patient needs for transcendence, legacy work, and existential resolution in the limited timeframe available. Future qualitative and case-based studies will be critical to further substantiate and refine this approach.”

2-Instead of being a structured assessment, the "Materials and Methods" section mostly reads like a narrative evaluation.  Indicate clearly that this is a narrative/conceptual review or provide clarification on if systematic procedures (databases searched, inclusion criteria) were applied.

R: An indication that this is a narrative/conceptual review was included in the Material and Methods Section:

“This analysis is based on a narrative/conceptual review of the scientific literature and provides essential context for the development of an integrative model that draws selectively from each tradition to address the complex psychological and existential needs of individuals in palliative care.”

3-The suggested model and the Montreal model for TRD are compared in the discussion.  This is useful, but to better contextualize the contribution, the comparison might be extended to other psychedelic protocols (such as psilocybin-assisted therapy in end-of-life care).

R: To address this issue, the following paragraph was added in the Discussion section:

“Psilocybin-assisted psychotherapy (PAP) protocols share core principles of safety, preparation, and integration, while varying in dose, format, and therapeutic focus. Therapeutic doses for depression typically center around 25 mg, with some studies using weight-adjusted or preparatory lower-dose sessions, which may themselves have antidepressant effects. Most protocols employ individualized therapy, emphasizing exploration of depressive thoughts and behaviors, whereas in long-term illness populations, therapy may include group sessions and focus on illness-related distress, relational support, and existential meaning. Across all models, psychotherapy is essential not only for safety during dosing but for guiding insight, reframing symptoms, and supporting long-term psychological and relational growth.”

4-The implementation of this strategy in oncology and palliative care settings presents logistical, regulatory, and resource problems that should be more clearly addressed in the research (e.g., staff training, legal access to ketamine, cost, cultural acceptance).

R: To address this issue, the following statement was added to the Discussion section:

“Despite its promise, the integration of ketamine-assisted psychotherapy (KAP) into palliative care and public health systems faces substantial challenges, including high drug costs, regulatory hurdles, infrastructure demands, and the need for systematic training of multidisciplinary teams. Beyond technical expertise, therapists must cultivate empathetic presence, trust-building, spiritual intelligence, and ethical integrity to guide patients through altered states of consciousness and support the integration of insights into coping and existential frameworks. Demonstrating cost-effectiveness and real-world feasibility will be essential for policymakers, but equally critical is the recognition that therapist preparation extends beyond pharmacological knowledge to encompass experiential and relational competencies that ensure KAP can be delivered safely, ethically, and meaningfully to populations with high unmet needs”.

5-Although existential healing is emphasized as a crucial result, the article offers no precise metrics for success.  Add validated instruments for existential distress, meaning, or dignity (e.g., Patient Dignity Inventory, FACIT-Sp).

R: We included in the Results section the following text to address the basic metrics that could be used to assess the outcomes of the KAP Model:

“Selecting appropriate outcome measures is critical for evaluating the impact of ketamine-assisted psychotherapy (KAP) in palliative care populations, where psychological, existential, and spiritual dimensions of suffering are often intertwined. Evidence from related interventions, such as Dignity Therapy, highlights a range of validated instruments that could be integrated into KAP studies. Recommended measures include multi-dimensional scales that capture both subjective experience and functional impact, such as the Patient Dignity Inventory (PDI) for dignity-related distress, the FACIT-Sp Meaning/Peace subscale for spiritual well-being, and the Distress Scale II (DS-II) or Edmonton Symptom Assessment System – spiritual domain (ESAS-Sp) for existential distress. Short-term assessments immediately post-session can capture acute ketamine effects, while longitudinal follow-ups (1–4 weeks) are essential to evaluate sustained outcomes. Complementary qualitative approaches, including semi-structured interviews, provide nuanced insight into meaning-making, transcendence, and spiritual experiences that may not be fully captured by standardized scales. Integrating these quantitative and qualitative measures offers a comprehensive framework to assess both the psychological and existential efficacy of KAP in palliative care”.

Reviewer 4 Report

Comments and Suggestions for Authors

The article, titled "Unfolding States of Mind: A Dissociative-Psychedelic Model of Ketamine-Assisted Psychotherapy in Palliative Care," presents an innovative and relevant proposal in the field of palliative care by articulating neurobiological, psychological, and phenomenological evidence on the use of ketamine in an intensive and brief psychotherapeutic model aimed at people at the end of life.
The choice of this topic is justified by the observation that suffering in this phase goes far beyond physical symptoms, involving existential anguish, loss of meaning, fear of death, and emotional fragility. The literature reveals that conventional treatments, whether pharmacological or psychotherapeutic, are often insufficient to address these aspects, which legitimizes the search for alternatives that integrate rapid, safe, and humane interventions.
From a theoretical perspective, the article is consistent and well-structured. The authors clearly review ketamine's mechanisms of action, describing its role as an NMDA antagonist, its relationship with neuroplasticity, and its impact on brain networks such as the Default Mode Network and the Salience Network. This foundation is enriched by a phenomenological perspective that differentiates dissociative experiences from psychedelic experiences and links them to distinct levels of self, bodily, and narrative processing. Furthermore, the comparison between existing models of ketamine's clinical use—biomedical, psychotherapeutic, and psychedelic—prepares the reader to understand the hybrid proposal adapted to palliative care. This theoretical framework is one of the work's strengths, as it combines findings from neuroscience with reflections from existential psychology and emerging clinical practices.
The model presented is pragmatic and sensitive to the limitations of the target audience. It is structured into preparation phases, two ketamine sessions—the first at a low dose, aimed at milder dissociative experiences, and the second at a moderate dose, with psychedelic potential—and integration phases that aim to consolidate meanings and align experiences with previously established care goals. The proposal emphasizes the therapeutic context, the "set and setting," the use of music, blindfolds, and the continuous presence of clinical support, aspects that denote a legitimate concern for psychological safety and the quality of the experience. The article also reinforces the importance of discussing care goals, aligning the intervention with essential palliative medicine practices, such as the emphasis on dignity, autonomy, and meaning in life.
However, despite its originality and relevance, the study has limitations that need to be analyzed, considered, and revised:
- It is a narrative, non-systematic review, which leaves room for selection bias in the selection of references: the method should be reviewed;
- The proposed model has not yet been empirically tested in palliative care populations, meaning there is no data to confirm its practical feasibility, safety, or efficacy in this specific population;
- The authors' arguments stem from extrapolations of studies conducted on people with resistant depression or drug addiction, who have distinct contexts in terms of clinical profile, prognosis, and vulnerability.
- Although the authors mention ethical issues, they do not sufficiently delve into critical aspects such as obtaining informed consent from fragile people in palliative care, the preparation of multidisciplinary teams, the necessary infrastructure, and implementation costs;
- The tone of the article, at times, reveals a high level of optimism regarding the transformative potential of ketamine, without discussing in depth the risks of disorganizing experiences or frustration in people at the end of life.
- In summary, the article constitutes a valuable contribution to the contemporary debate on psychedelic interventions in palliative care. Its merit lies in proposing an innovative conceptual model that integrates biological, psychological, and existential dimensions and proves appropriate for the reality of people with limited life expectancy and great fragility. However, because it is still a theoretical proposal without clinical validation, the text should be read as a starting point rather than a definitive guide for healthcare practice, which is why it deserves revision.

Author Response

1- It is a narrative, non-systematic review, which leaves room for selection bias in the selection of references: the method should be reviewed;

R: An indication that this is a narrative/conceptual review was included in the Material and Methods Section:

“This analysis is based on a narrative/conceptual review of the scientific literature and provides essential context for the development of an integrative model that draws selectively from each tradition to address the complex psychological and existential needs of individuals in palliative care.”

2- The proposed model has not yet been empirically tested in palliative care populations, meaning there is no data to confirm its practical feasibility, safety, or efficacy in this specific population;

R: An indication that feasibility, safety, or efficacy in this specific population should be tested in future studies was included in the Discussion section:

“Ongoing clinical trials are beginning to explore the feasibility and impact of ketamine-based interventions in palliative care. Current studies range from intranasal administration for depression and anxiety in early-stage cancer patients, to ketamine-assisted psychotherapy for terminally ill individuals facing death-related distress, as well as trials assessing subcutaneous, oral, and combined approaches for pain and mood symptoms in advanced illness. While still preliminary, these investigations emphasize feasibility, safety, and integration of caregivers, and they hold promise for expanding therapeutic options in addressing the complex psychological, existential, and physical dimensions of suffering at the end of life.”

3- The authors' arguments stem from extrapolations of studies conducted on people with resistant depression or drug addiction, who have distinct contexts in terms of clinical profile, prognosis, and vulnerability.

R: These considerations were added to the discussion section so that the proposed model should be viewed with caution, in light of the scarcity of data specifically for the palliative care population:

“It is important to highlight that the rationale for a two-session KAP model in palliative care draws from both empirical findings in adjacent populations and the unique temporal constraints of terminal illness. While single-dose ketamine has demonstrated rapid antidepressant effects in treatment-resistant depression, individuals facing life-limiting conditions often require an approach that not only alleviates mood symptoms but also addresses existential, spiritual, and relational dimensions of suffering. In this context, a time-limited, two-session format represents a pragmatic compromise: intensive enough to facilitate meaningful psychotherapeutic work within the short prognosis typical of palliative care, yet sufficiently streamlined to remain feasible in clinical practice. Although direct evidence in palliative populations is currently lacking, this model represents a theoretically grounded extrapolation, one that prioritizes patient needs for transcendence, legacy work, and existential resolution in the limited timeframe available. Future qualitative and case-based studies will be critical to further substantiate and refine this approach”.

4- Although the authors mention ethical issues, they do not sufficiently delve into critical aspects such as obtaining informed consent from fragile people in palliative care, the preparation of multidisciplinary teams, the necessary infrastructure, and implementation costs;

R: To address these issues, the following statements were included:

“KAP in palliative care raises important ethical considerations that must be addressed to ensure patient-centered and responsible practice. These include respecting patient autonomy and informed consent, particularly given the altered states of consciousness induced by ketamine, and carefully weighing potential benefits against risks in a population often characterized by vulnerability and frailty. Ethical implementation also requires transparency regarding therapeutic goals, the limits of evidence, and potential side effects, as well as sensitivity to cultural and spiritual dimensions of end-of-life care. Incorporating ongoing ethical reflection, multidisciplinary consultation, and robust documentation can help safeguard patients’ dignity, promote trust, and support equitable access to KAP within palliative care frameworks.”

5- The tone of the article, at times, reveals a high level of optimism regarding the transformative potential of ketamine, without discussing in depth the risks of disorganizing experiences or frustration in people at the end of life.

R: To address this issue, a new subsection was included:

“3.7. Contraindications, adverse events, and monitoring requirements

Ketamine, when administered at subanesthetic doses in palliative care, is generally well tolerated, but several adverse effects may occur. Short-term effects include dissociation, psychotomimetic symptoms, transient anxiety, blurred vision, dizziness, headache, nausea, and vomiting. Cardiovascular effects are common, with temporary increases in systolic and diastolic blood pressure (typically 8–20 mmHg and 8–13 mmHg, respectively) and heart rate, peaking within 30–60 minutes of infusion and resolving within 1–2 hours. Less frequent effects include chest discomfort or palpitations. Patients with psychotic features, uncontrolled hypertension, or significant cardiovascular disease are generally considered contraindicated due to heightened risk.

Longer-term adverse effects, such as abuse potential, neurotoxicity, bladder dysfunction, and hepatotoxicity, have primarily been observed in the context of recreational or chronic misuse and have not been reported in therapeutic settings. Nevertheless, careful patient selection, pre-infusion screening for psychiatric, cardiovascular, and substance use conditions, individualized dosing, and structured follow-up remain essential. Monitoring requirements include continuous observation of cardiovascular parameters during and shortly after infusion, readiness to manage acute hemodynamic changes, and structured assessment of psychological responses to optimize safety and therapeutic outcomes in palliative care population”.

Round 2

Reviewer 2 Report

Comments and Suggestions for Authors

The authors have addressed all the comments and improved the manuscript now.

Author Response

The authors would like to thank the reviewer for their valuable time and insightful comments, which helped to improve the clarity and quality of this manuscript.

Reviewer 3 Report

Comments and Suggestions for Authors

The authors have properly address all the comments that was raised during the first round of review. Hence, I would like to accept the manuscripts in its current form. 

Author Response

(The authors gave the same response as above.)

Reviewer 4 Report

Comments and Suggestions for Authors

Thank you for the opportunity to review the proposed manuscript again.

As mentioned in the previous review, authors should review the methodology presented in the article and ensure it meets the requirements of international guidelines.

Author Response

1- As mentioned in the previous review, authors should review the methodology presented in the article and ensure it meets the requirements of international guidelines.

R: To address this issue, the following paragraph was added in the Methods section:

“This study followed a narrative review design aimed at providing an interpretative and integrative synthesis of the literature on KAP in palliative care. Guided by an interpretivist perspective, the review sought to describe current knowledge, theoretical frameworks, and clinical approaches related to KAP in end-of-life contexts.

An iterative and flexible search strategy was conducted across major scientific databases and relevant grey literature, focusing on studies, conceptual papers, and clinical reports addressing the therapeutic use of ketamine for psychological, existential, or spiritual distress in palliative care. Sources were selected based on conceptual and thematic relevance rather than strict inclusion criteria.

The analysis involved identifying recurrent themes, models, and gaps within the literature, integrating empirical findings with theoretical perspectives from psychotherapy, psychedelic science, and palliative medicine. Reflexivity was maintained throughout the process, recognizing the influence of the authors’ clinical and theoretical background on interpretation.”